# VIBE: Topic-Driven Temporal Adaptation for Twitter Classification

**Yuji Zhang**[1*] **Jing Li**[1,2†] **Wenjie Li**[1,2]

[1]Department of Computing, The Hong Kong Polytechnic University, HKSAR, China
[2]Research Centre on Data Science & Artificial Intelligence
yu-ji.zhang@connect.polyu.hk
{jing-amelia.li, wenjie.li}@polyu.edu.hk

## Abstract

Language features are evolving in real-world social media, resulting in the deteriorating performance of text classification in dynamics. To address this challenge, we study *temporal adaptation*, where models trained on past data are tested in the future. Most prior work focused on continued pretraining or knowledge updating, which may compromise their performance on noisy social media data. To tackle this issue, we reflect feature change via modeling latent topic evolution and propose a novel model, **VIBE**: **V**ariational **I**nformation **B**ottleneck for **E**volutions. Concretely, we first employ two *Information Bottleneck* (IB) regularizers to distinguish past and future topics. Then, the distinguished topics work as adaptive features via multi-task training with timestamp and class-label prediction. In adaptive learning, VIBE utilizes retrieved unlabeled data from online streams created posterior to training data time. Substantial Twitter experiments on three classification tasks show that our model, with only 3% of data, significantly outperforms previous state-of-the-art continued-pretraining methods.

## 1 Introduction

Our ever-changing world leads to the continuous data distribution shifting in social media. As a result, language features, formed by word patterns, will consequently evolve over time. Therefore, many text classification models, including the state-of-the-art ones based on pretraining, demonstrate compromised results facing shifted fea-

---

*Code will come later at https://github.com/CelestineZYJ/VIBE-Temporal-Adaptation. For technical questions, contact <yu-ji.zhang@connect.polyu.hk>.

†This work is supported by the NSFC Young Scientists Fund (Project No. 62006203), a grant from the Research Grants Council of the Hong Kong Special Administrative Region, China (Project No. PolyU/25200821), the Innovation and Technology Fund (Project No. PRP/047/22FX), and PolyU Internal Fund from RC-DSAI (Project No. 1-CE1E). Jing Li is corresponding author. The authors would like to thank the anonymous reviewers from EMNLP 2023 for their insightful suggestions of various aspects of this work.

---

| Evolving Topics Over Time |
|---|
| $T + 0$: *Covid, NoMask, ReturnToWork, Dread* |
| $T + 1$: *WearAMask, LockDown, ImmuneCompromise* |
| $T + 2$: *Symptom, PanicBuy, ImWithFauci, TrumpKillUs* |
| $T + 3$: *Therapy, TrumpisANationalDisgrace, LyingTrump* |
| $T + 4$: *RedStates, TrumpLandSlid, TrumpLiesAmericans* |

Figure 1: Evolving topics in COVID-19 *Stance* dataset over time. $T+\tau$ ($\tau$=0, 1, ..., 4) is in chronological order.

tures (Hendrycks et al., 2020), when there is a time gap between training and test data. Röttger and Pierrehumbert (2021) and Luu et al. (2022) then defined a *temporal adaptation* task to adapt a model trained on the past data to the shifted future data.

To tackle this task, Röttger and Pierrehumbert (2021) and Luu et al. (2022) apply continued pretraining on the temporally-updated data to catch the distribution shift yet observe limited enhancement with noisy contexts, meanwhile relying heavily on computational resources. Others (Shang et al., 2022; Li et al., 2022; Chen et al., 2022) resort to knowledge-based methods, which require high-quality data unavailable on social media.

Given these concerns, how shall we effectively approach temporal adaptation in noisy social media contexts? Inspired by the success of latent topics in many social media tasks (Zhang et al., 2021; Pathak et al., 2019), our solution is to explore latent topic evolution in representing feature change (Zhang and Li, 2022). To better illustrate our idea, we exemplify the COVID-19 stance detection dataset (Glandt et al., 2021). Here we employ the Neural Topic Model (NTM) (Miao et al., 2017) to capture the latent topics in five sequential periods and show the top topic words by likelihoods in Figure 1. As can be seen from the topic evolution, users changed discussion points rapidly over time, where they shifted focus from concerns to the virus itself (e.g., "*Immune Compromise*", "*Symptom*") to the disappointment to the former US President Trump (e.g., "*Trump Land Slid*", "*Lying Trump*").

From this example, we can learn that topics can help characterize evolving social media environ-

ments, which motivates us to propose topic-driven temporal adaptation. Specifically, we employ an NTM for featuring latent topics, which is under the control of *Information Bottleneck* (IB) regularizers to distinguish *past-exclusive*, *future-exclusive*, and *time-invariant* latent topics to represent evolution. Here the time-invariant topic is to encode the task-relevant semantics unchanged over time; the past- and future-exclusive latent topics capture the time-variant topics by modeling the data from the past and future. To further align the topic evolution to a specific classification task, we leverage multi-task training to infer timestamps and class labels jointly. Our model is then named **VIBE**, short for **V**ariational **I**nformation **B**ottleneck for **E**volutions.

To study how VIBE functions on social media data, we utilize the data temporally posterior to training data to learn adaptation (henceforth *adaptive data*). Here we consider two scenarios. First, in an ideal circumstance, the data between training and test time from the original dataset can serve as the *golden adaptive data*. Second, when it is inaccessible in a realistic setup, we retrieve relevant online streams as the *retrieved adaptive data*.

For experiments, we conduct a temporal adaptation setup on three text classification tasks with data from Twitter platform and draw the following results. First, the main comparison shows that VIBE outperforms the previous state-of-the-art (SOTA) models with either golden or retrieved adaptive data. For example, VIBE brings a relative increase of 5.5% over the baseline for all datasets, using only 3% of the continued pretraining data. Then, an ablation study exhibits the positive contribution of VIBE's sub-modules, and VIBE is insensitive to retrieved adaptive data quality. Next, we examine the effects of adaptive data quantity on VIBE, and the results show that VIBE relies much less on data volume compared to continued pretraining. At last, we present a case study to interpret how VIBE learns the evolving topic patterns for adaption.

To sum up, we present three-fold contributions:

• *To the best of our knowledge, we are the first to explore how to model latent topic evolution as temporal adaptive features from noisy contexts.*

• *We propose a novel model VIBE coupling the information bottleneck principle and neural topic model for temporal adaptation in social media.*

• *Substantial Experiments show that VIBE significantly outperforms previous SOTA models, and is insensitive to adaptive data quantity.*

## 2 Related Work

### 2.1 Temporal Adaptation

Our task is in line with temporal adaptation, a sub-task of domain adaptation (DA). While prior DA studies focus on domain gaps exhibiting intermittent change, temporal distribution shift usually happens step by step, forming a successive process. Liu et al. (2020a) showed the impracticality of applying traditional DA on temporal adaptation because the former aims to eliminate the distribution shift (Xie et al., 2021; Shen et al., 2021; Liu et al., 2020b; Hendrycks et al., 2020; Chen et al., 2021) while the adaptive objective for the latter is to track the shifts. To track the shifts, Huang and Paul (2019) utilized more labeled data to adapt, resulting in high costs in the ever-evolving environments.

Unsupervised temporal adaptation was therefore advanced in two lines. One is based on continued pretraining on large-scale future data (Röttger and Pierrehumbert, 2021; Luu et al., 2022; Huang et al., 2022), relying on extensive computational resources whereas interior with a noisy context. The other adopted external knowledge resources (Shang et al., 2022; Li et al., 2022; Chen et al., 2022, 2023; Song and King, 2022; Liang et al., 2022), requiring high-quality data unavailable in social media. Differently, we feature evolving latent topics with less reliance on data quality and quantity.

### 2.2 Topic Modeling for Social Media

A topic model is a statistical model for discovering abstract topics occurring in a collection of texts (Miao et al., 2017), whose potential for dealing with dynamics comes from its capability to cluster texts exhibiting similar word statistics. Many previous studies (Bak et al., 2014; Naskar et al., 2016; Srivatsan et al., 2018) have demonstrated the effectiveness of topic models to encode social media texts. Furthermore, Saha and Sindhwani (2012); Pathak et al. (2019); Balasubramaniam et al. (2021) analyzed the evolving topic semantics via topic matrix factorization. Nevertheless, none of the above studies explore how latent topics help social media temporal adaptation, which is a gap we mitigate.

## 3 Preliminaries

To prepare readers for understanding VIBE's model design (to appear in §4), we present a preliminary section here with the problem formulation in §3.1, information bottleneck principle (VIBE's theoretical basis) in §3.2, and the model overview in §3.3.

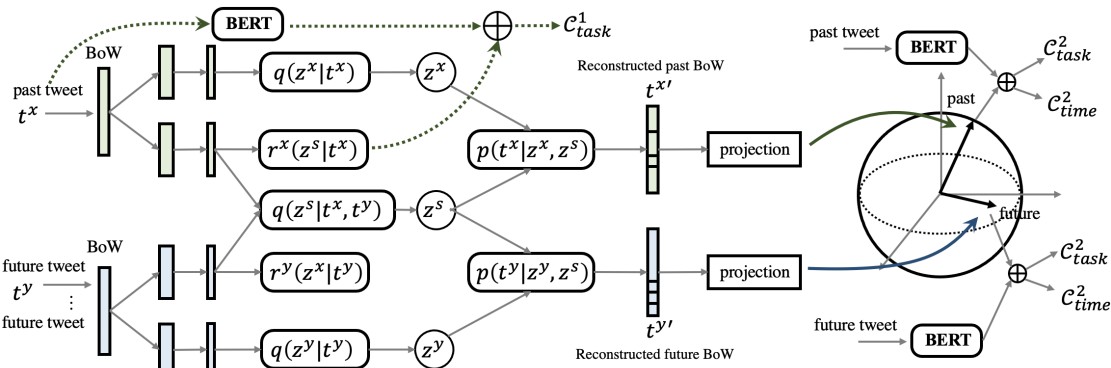

Figure 2: The framework of our model VIBE. Taking past and future tweets $t^x$, $t^y$ as the input, VIBE derives the latent topics they conditioned on, as past-exclusive, future-exclusive, and time-invariant topics $z^x$, $z^y$, and $z^s$. $C^1_{task}$ is a MLP for first stage training on predicting class label. $C^2_{task}$ and $C^2_{time}$ are for second stage multi-task training.

## 3.1 Problem Formulation

We explore temporal adaptation in text classification on Twitter, where an input tweet sample denotes $t$, and its class label $c$. Following Luu et al. (2022), the training data is from the past period $X$ with labels (denoted $D_X = (t^x, c^x)$); In contrast, adaptive and test data are unlabeled and from the future period $Y$ (denoted $D_Y = (t^y, \_)$). Our goal here is to enable *VIBE to infer accurate future label $c^y$ via exploring the topic evolution from $t^x$ to $t^y$*.

To that end, VIBE explicitly captures time-invariant, past-exclusive, and future-exclusive topics, respectively denoted $z^s$, $z^x$, and $z^y$, which are extracted and differentiated from the past and future tweets, $t^x$ and $t^y$. $z^x$ represents the past and can only be learned from $t^x$, not from $t^y$; likewise, $z^y$ reflects the future and exclusively originates from $t^y$ without information from $t^x$. The shared part of $t^x$ and $t^y$ is modeled by $z^s$, unchanged over time yet helpful in indicating class labels.

For adaption, VIBE should be fed with the adaptive data, $N$ samples of unlabeled $t^y$ from the future time $Y$ (from either golden or retrieved adaptive data) are selected on relevance to each past tweet $t^x$ (from training data). Taking $t^x$ (past) and $t^y$ (future) as input, VIBE adopts the information bottleneck principle (introduced in §3.2) to distinguish topics $z^s$, $z^x$, and $z^y$, which will be detailed in §4.2.

## 3.2 Information Bottleneck Principle

Information Bottleneck (IB) Principal (TISHBY, 1999; Tishby and Zaslavsky, 2015) was designed for pursuing a maximally informative representation with a tradeoff for general predictive ability (Alemi et al., 2017). Its objective function is:

$$\mathcal{L}_{IB} = I(Z; X) - I(Z; Y). \quad (1)$$

Here $I(\cdot; \cdot)$ is the mutual information of two random variables. Minimizing $\mathcal{L}_{IB}$ enables $Z$ to be maximally expressive for $Y$ with a tradeoff of minimally relying on $X$ (Lin et al., 2020; Wu et al., 2020). In VIBE, take time-invariant topics $z^s$ as an example, for the past tweets $t^x$, the IB regularizers force the time-invariant topics $z^s$ to be expressive for $t^x$, meanwhile penalizing disturbance from future tweets $t^y$. For the future tweets $t^y$, vice versa. Thus, the representation of $z^s$ is distinguished from time-variant $z^x$ and $z^y$. For tractability of mutual information, the variational approximation (Alemi et al., 2017) is adopted to optimize the IB objective function by constructing a lower bound.

## 3.3 Model Overview

Based on §3.1 and 3.2, we then brief an overview of VIBE before detailing its design in §4.

For modeling latent topics, we adopt NTM (the vanilla NTM is elaborated in A.1) to take bag-of-word (BoW) vectors of past tweets $t^x$ and future tweets $t^y$ as input to derive past-exclusive, future-exclusive, and time-invariant latent topic variables. During the learning process, the two BoW vectors are first encoded into latent topics $z^s$, $z^x$, and $z^y$, then decoded to reconstruct the BoW vectors. Their disentanglement are handled by the IB regularizers, which will be elaborated in §4.2.

In addition, to encode more useful features, $z^s$ is endowed with task-specific semantics by using $(z^s \,|\, t^x)$ to infer the class label $c^x$ of past tweet $t^x$. We then employ the trained classifier $C^1_{task}$ to pseudo-label the future tweets as $c^{y'}$. With the reconstructed BoW of $t^x$ and $t^y$, carrying the temporal topic patterns from the input, we employ multi-task training to infer timestamps and class labels to align topic evolution features to classification.

## 4 VIBE Model

Here §4.1 first describes how NTM explores the distribution of past and future tweets. Then, we discuss the design of time-oriented (§4.2) and task-oriented training (§4.3) to feature the time-invariant and time-variant latent topic variables by IB. Next, §4.4 presents how to jointly optimize the NTM and classifier, followed by the application of the topic evolution features for final prediction in §4.5.

### 4.1 NTM with Multiple Latent Variables

As discussed in §3.1, NTM is used to capture the latent topic evolution by generating the past and future tweets $t^x$, $t^y$ based on latent topics. We assume that paired past-future tweet samples $(t^x, t^y)$ come from the joint distribution $(t^x, t^y) \sim p_D(t^x, t^y)$. The distribution $p_D(t^x, t^y)$ is jointly conditioned on time-variant topics (past-exclusive topics $z^x$ and future-exclusive topics $z^y$) and time-invariant topics $z^s$. To learn the latent topics, we pursue the maximization of the marginal likelihood (Diederik et al., 2014) of the distribution:

$$p_\theta(t^x, t^y) = \int dz^x dz^s dz^y p_{\theta_X}(t^x | z^x, z^s)$$
$$p_{\theta_Y}(t^y | z^y, z^s) p(z^x) p(z^s) p(z^y) \qquad (2)$$

Here the involved parameter $\theta = \{\theta_X, \theta_Y\}$ models the conditional distributions; our training objective is to maximize the generative model $p_\theta(t^x, t^y)$. To allow tractable optimization in training, we substitue $p_\theta(z^x, z^s, z^y | t^x, t^y)$ as the approximated posterior $q_\phi(z^x, z^s, z^y | t^x, t^y)$ based on the variational inference method (Diederik et al., 2014) and the approximated posterior is factorized into:

$$q_\phi(z^x, z^s, z^y | t^x, t^y) = q_{\phi_X}(z^s | t^x) q_{\phi_S}(z^s | t^x, t^y) q_{\phi_Y}(z^y | t^y) \qquad (3)$$

where $\phi = \phi_X, \phi_S, \phi_Y$ denotes the entire encoder parameter — $q_{\phi_X}$ and $q_{\phi_Y}$ reflect the past-exclusive and future-exclusive topics to X (past time) and Y (future time), and $q_{\phi_S}$ represents time-invariant topics. For inference, we derive evidence lower bound (ELBO) of $p_\theta(t^x, t^y)$ (More in A.2):

$$\log p(t^x, t^x) \geq \mathbb{E}_{q(z^x|t^y)q(z^s|t^x,t^y)} \left[ \log p(t^x | z^x, z^s) \right]$$
$$+ \mathbb{E}_{q(z^y|t^y)q(z^s|t^x,t^y)} \left[ \log p(t^y | z^y, z^s) \right]$$
$$- D_{KL} \left[ q(z^x|t^x) \| p(z^x) \right] - D_{KL} \left[ q(z^y|t^y) \| p(z^y) \right]$$
$$- D_{KL} \left[ q(z^s|t^x, t^y) \| p(z^s) \right] \qquad (4)$$

However, three encoders $q(z^x | t^x)$, $q(z^s | t^x, t^y)$, and $q(z^y | t^y)$ possibly encodes $z^x$, $z^y$ and $z^s$ with substantial overlaps in their captured semantics. To

differentiate them for modeling topic evolution, we apply IB regularizers to force disentanglement of topics $z^x$, $z^y$, and $z^s$ following Hwang et al. (2020) (see §4.2). Additionally, we endow latent topics $z^s$ with time-invariant task-specific semantics by training $z^s | t^x$ to infer $t^x$'s label $c^x$ (see §4.3).

### 4.2 Time-Oriented Training for NTM

In the following, we elaborate on how the IB regularizers control NTM to learn the disentangled past-, future-exclusive and time-invariant topics.

**Time-Invariant IB Regularizer.** This regularizer is designed to capture time-invariant topics $z^s$ shared by the past and future tweets, whose design follows the interaction information theory (McGill, 1954). Interaction information refers to generalizing mutual information to three or more random variables, representing the amount of shared information between them. Concretely, we denote the shared information between past time $X$ and future time $Y$ as $Z^S$ and present the formulation for interaction information $I(X; Y; Z^S)$ as follows:

$$I(X; Y; Z^S) = I(X; Z^S) - I(X; Z^S | Y) \qquad (5)$$

$$I(X; Y; Z^S) = I(Y; Z^S) - I(Y; Z^S | X) \qquad (6)$$

The above two equations represent $I(X; Y; Z^S)$ through a symmetrical lens. As can be seen, in Eq. 5, we maximize the interaction information by maximizing the first term $I(X; Z^S)$ to make $Z^S$ expressive for past time $X$ and minimizing the second term $I(X; Z^S | Y)$ to penalize the disturbance from future time $Y$. Symmetrically, Eq. 6 encourages $Z^S$ to be expressive for future time $Y$ and avoid being affected by past time $X$. Optimizing these two functions simultaneously would allow $Z^S$ to be time-invariantly expressive of $X$ and $Y$.

**Time-Variant IB Regularizer.** Our goal is to minimize the mutual information $I(Z^X; Z^S)$ and $I(Z^Y; Z^S)$, thus past-, future-exclusive topics $z^x$, $z^y$ are statistically independent of the time-invariant topics $z^s$, and vice versa. Here we only present the formulation for the past time period $X$, and that for future time $Y$ can be inferred anomalously. The mutual information of $Z^X$ and $Z^S$ is:

$$I(Z^X; Z^S) = -I(X; Z^X, Z^S) + I(X; Z^X) + I(X; Z^S). \qquad (7)$$

$I(X; Z^X, Z^S)$ is maximized to force $Z^S$ and $Z^X$ to be jointly informative of past time $X$, while penalizes last two terms to avoid either $Z^S$ or $Z^X$ to be individually expressive for $X$ (more in A.2).

**Joint Regularization for IB Regularizers.** Time-variant and time-invariant representations are jointly regularized. We combine Eq. 5 and 7 for time X's regularization, and time Y's is analogous.

$$
\begin{aligned}
\max_q \quad & I(X;Y;Z^S) - I(Z^X;Z^S) \\
& = I(X;Z^X,Z^S) - I(X;Z^X) - I(X;Z^S|Y).
\end{aligned}
$$
(8)
$$
\begin{aligned}
\max_q \quad & I(X;Y;Z^S) - I(Z^Y;Z^S) \\
& = I(Y;Z^Y,Z^S) - I(Y;Z^Y) - I(Y;Z^S|Y).
\end{aligned}
$$
(9)

**Tractable Optimization for IB Regularizers.** Since there are intractable integrals like unknown distribution $p_D(t^x, t^y)$ in mutual information terms mentioned above, we maximize generative distributions' lower bounds for traceability concerns. More details for Eq. 10-Eq. 12 are discussed in A.2.

For $I(X;Z^X;Z^Y)$, we derive its lower bound with the generative distribution $p(t^x \mid z^x, z^s)$:

$$
\begin{aligned}
I\left(X;Z^X,Z^S\right) &= \mathbb{E}_{q(z^x,z^s|t^x)p_D(t^x)}\left[\log \frac{q\left(t^x|z^x,z^s\right)}{p_D(t^x)}\right] \\
&\geq H(X) + \mathbb{E}_{p_D(t^x,t^y)q(z^x|t^x)q(z^s|t^x,t^y)}\left[\log p\left(t^x|z^x,z^s\right)\right]
\end{aligned}
$$
(10)

For intractable $-I(X;Z^X)$ (unknow distribution $p_D(t^x)$), the generative distribution $p(z^x)$ is adopted as the standard Gaussian, known as the Variational Information Bottleneck (Tishby and Zaslavsky, 2015). Its lower bound goes as follows:

$$
I(X;Z^X) \geq \mathbb{E}_{P_D(t^x)}[D_{KL}[q(z^x|t^x)\|p(z^s)]]
$$
(11)

Likewise, the lower bound for $-I(X;Z^S|Y)$ is:

$$
\begin{aligned}
-I(X;Z^S|Y) &= -\mathbb{E}_{p_D(t^x,t^y)q(z^s|t^x,t^y)}[\log \frac{q(z^s|t^x,t^y)}{q(z^s|t^y)}] \\
&\geq -\mathbb{E}_{p_D(t^x,t^y)}\left[D_{KL}\left[q\left(z^s|t^x,t^y\right)\|r^y\left(z^s|t^y\right)\right]\right]
\end{aligned}
$$
(12)

Substituting Eq. 10-12 to Eq. 8-9, we derive the lower bound for disentangling past time $X$ and future time $Y$. For joint regularization on disentanglement and the maximum likelihood objective in Eq. 4, we combine them for joint maximization:

$$
\begin{aligned}
\max_{p,q} \mathbb{E}_{q(z^x,z^s,z^y,t^x,t^y)}&[\log \frac{p(t^x,t^y,z^x,z^s,z^y)}{q(z^x,z^s,z^y|t^x,t^y)}] \\
&+ \lambda(2 \cdot I(X;Y;Z^S) - I(Z^X;Z^S) - I(Z^Y;Z^S)) \\
&\geq \max_{p,q,r}(1+\lambda) \cdot \mathbb{E}_{p_D(t^x,t^y)}[ELBO(p,q)] \\
&+ \lambda \cdot \mathbb{E}_{p_D(t^x,t^y)}[D_{KL}[q(z^s|t^x,t^y)\|p(z^s)]] \\
&- \lambda \cdot \mathbb{E}_{p_D(t^x,t^y)}[D_{KL}[q(z^s|t^x,t^y)\|r^y(z^s|t^y)] \\
&+ D_{KL}[q(z^s|t^x,t^y)\|r^x(z^s|t^x)]]
\end{aligned}
$$
(13)

This objective is denoted as $\mathcal{L}_{ntm}$ (the NTM loss).

### 4.3 Task-Oriented Training for NTM

We have discussed controlling the latent topics $z^s$ to represent time-invariant features shared by past and future. Then we dive deep into how to inject the task-specific features to $z^s$ by inferring class labels. Specifically, we adopt the posterior latent topics $z^s \mid t^x$ (instead of directly using $z^s$) to leverage the task-specific factor reflected by the class label $c^x$ and its input $t^x$. Here we concatenate tweet $t^x$'s BERT embedding $b^{t^x}$ and the topic features $r^x(z^s \mid t^x)$ yielded by NTM encoder ($r^x(\cdot)$, shown in Eq. 12) for predicting the class label $c^x$. At the output layer, the learned classification features are mapped to a label $\hat{c}^{t^x}$ with the formula below:

$$
\hat{c}^{t^x} = f_{out}(\mathbf{W}_{out} \cdot \mathbf{u}_x + \mathbf{b}_{out})
$$
(14)

$f_{out}(\cdot)$ is the activation function for classification output (e.g., softmax). $\mathbf{W}_{out}$ and $\mathbf{b}_{out}$ are learnable parameters for training. $\mathbf{u}_x$ couples the BERT-encoded semantics ($\mathbf{b}^{t^x}$) and the implicit task-specific topics via a multi-layer perceptron (MLP):

$$
\mathbf{u}_x = f_{MLP}(\mathbf{W}_{MLP}[\mathbf{b}^{t^x};\mathbf{r}^x(z^s|t^x)] + \mathbf{b}_{MLP})
$$
(15)

### 4.4 Joint Training for NTM and Classification

To coordinate NTM and classifier training, we combine classification loss and the NTM reconstruction loss together. For classification, the loss $\mathcal{L}_{past}$ is based on cross-entropy (trained on labeled past data) Adding $\mathcal{L}_{NTM}$ (Eq. 13) to $\mathcal{L}_{past}$, we get the joint optimization loss function for training the temporally-adaptive classifier $C_{task}^1$, whose joint-training loss is the weighted sum of classification (task learning) and NTM (topic evolution) losses:

$$
\mathcal{L} = \mathcal{L}_{past} + \mu \cdot \mathcal{L}_{ntm}
$$
(16)

Here $\mu$ trades off the effects of task and topic evolution learning. In this way, their parameters are updated together. After the training, we employ the trained $C_{task}^1$ to pseudo label future tweets $t^y$ from adaptive data to form $D_Y = (t^x, c^{y\prime})$ for the next stage training, which will be discussed in §4.5.

### 4.5 Topic Space Projection

In the original design of NTM decoding (Miao et al., 2017), $t^x$ and $t^y$ are reconstructed based on a ruthless vocabulary dictionary. We tailor-make it to map the reconstructed $t^x$ and $t^y$ into a sphere space to align tweets with similar semantics to exhibit closer distance in topic space. Meanwhile, it is conducted via multi-task learning on predicting timestamps and class labels to inject time-awareness senses into the classification training:

$$
\hat{y}^t = f_{y_{out}}(\mathbf{W}_{y_{out}} \cdot \mathbf{u}_t + \mathbf{b}_{y_{out}})
$$
(17)

| Dataset | Scale | Adaptive | Time Span |
|---------|-------|----------|-----------|
| *Stance* | 7,122 | 35,610 | 2020/Feb-2020/Sep |
| *Hate* | 8,773 | 43,860 | 2018/Oct-2020/July |
| *Hash* | 22,415 | 65,820 | 2011/Sep-2011/Dec |

Table 1: Data statistics. The "Scale" column shows the dataset's tweet number and the "Adaptive" column shows the number of tweets as retrieved adaptive data.

$$\hat{T}^t = f_{T_{out}}(\mathbf{W}_{T_{out}} \cdot \mathbf{u}_t + \mathbf{b}_{T_{out}}) \qquad (18)$$

$f(\cdot)$ is the activation function for classification output. $\mathbf{W}$ and $\mathbf{b}$ are learnable parameters for training. $\mathbf{u}_t$ concatenates tweet $t$'s BERT embedding $b^t$ and its reconstructed BoW vector for class label $y^t$ and timestamp $T^t$ prediction via a MLP. The two classifiers are named $C^2_{task}$ and $C^2_{time}$. The cross-entropy losses for $y^t$ and $T^t$ prediction are denoted as $\mathcal{L}_{task}$ and $\mathcal{L}_{time}$. We optimize their joint loss as follows:

$$\mathcal{L}_{sphere} = \mathcal{L}_{task} + \mathcal{L}_{time} \qquad (19)$$

## 5 Experimental Setup and Data Analysis

In the following, we introduce data (including its analysis) and baselines for the experimental setup.

**Data Collection.** As displayed in Table 1, we experiment with VIBE on three datasets for stance detection (*Stance*) (Glandt et al., 2021), hate speech (*Hate*) (Mathew et al., 2021), and hashtag (*Hash*) prediction. Their data is all from Twitter, and each sample is a tweet with a classification label. We show more details of datasets in A.3.

**Data Setup.** For the Stance and Hate dataset, tweets were sorted by time order, where the earliest 50% samples are for training and the latest 30% for test. The middle 20% tweets posted between training and test time were adopted for validation (5% random samples) and the golden adaptive data (the rest 15%). For Hash dataset whose data is unevenly distributed temporally, we adopted the absolute temporal split: Sep and Oct data for training,

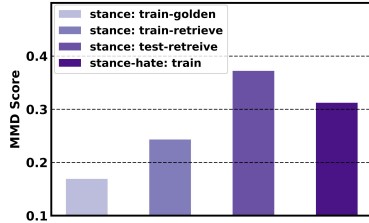

Figure 4: MMD score to quantify distribution shifts. The gap between the test and retrieved adaptive data is larger than the cross-task one between Stance and Hate.

Nov for adaptation and validation, Dec for the test. In our setup, training data (with labels) are from the past; adaptive and test data are from the future.

For gathering the retrieved adaptive data, we first analyzed the top 100 trendy words (excluding stopwords) in training data. Then, we searched their related tweets and sampled those posted in the same time period of test data (50 per day for each search word). Finally, a retrieval method (e.g., DPR) was adopted for ranking and selecting tweets with the top-$N$ highest similarity to each training sample to form the adaptive data.

**Data Analysis.** As previously shown, Twitter usually challenges NLP models with its noisy data (Wang et al., 2019). We then exemplify the Stance dataset and provide analysis to quantify unique challenges from noisy data to temporal adaptation.

Following Gururangan et al. (2020)'s practice, we first analyze the vocabulary overlap of four subsets — training set, golden adaptive data (from the original dataset), retrieved adaptive data (external data retrieved from the wild), and test set. As shown in Figure 3, the vocabulary's inter-subset overlap ratio is low, around 40-50%, in contrast to 60-90% in non-Twitter data (Luu et al., 2022). It indicates the grand challenge of capturing temporal adaptive features on Twitter, echoing Huang and Paul (2019)'s findings that Twitter data tends to generate a vocabulary gap faster than other data.

| Model | Stance | Hate | Hash |
|-------|--------|------|------|
| ERM | $0.8014_{\pm 0.5673}$ | $0.6603_{\pm 0.7279}$ | $0.8153_{\pm 2.7880}$ |
| DANN | $0.7959_{\pm 0.4481}$ | $0.6674_{\pm 0.4639}$ | $0.8083_{\pm 3.2849}$ |
| Mixup | $0.7901_{\pm 0.6277}$ | $0.6580_{\pm 0.9159}$ | $0.7915_{\pm 2.5351}$ |
| MMD | $0.7836_{\pm 0.3799}$ | $0.6611_{\pm 0.6981}$ | $0.8114_{\pm 2.9935}$ |
| P+CFd | $0.8028_{\pm 0.5466}$ | $0.6708_{\pm 0.1637}$ | $0.8171_{\pm 3.6226}$ |
| UDALM | $\mathbf{0.8120}_{\pm 0.4542}$ | $\mathbf{0.6755}_{\pm 0.5250}$ | $\mathbf{0.8341}_{\pm 3.1724}$ |
| DPT | $\mathbf{0.8173}_{\pm 0.2308}$ | $\mathbf{0.6712}_{\pm 0.1731}$ | $\mathbf{0.8306}_{\pm 3.0942}$ |
| **VIBE** | $\mathbf{0.8368}_{\pm 0.1914}$ | $\mathbf{0.6900}_{\pm 0.3501}$ | $\mathbf{0.8551}_{\pm 2.4463}$ |

Table 2: Classification accuracy with golden adaptive data. We report mean scores±standard deviation (std.) in ten runs. The order of magnitude for std. is $10^{-5}$.

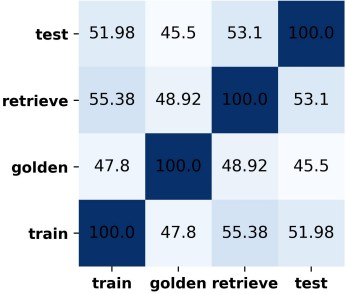

Figure 3: Pairwise vocabulary overlap (%): the vocabulary gathers the top 5K most frequent words (excluding stop words) from each subset.

| Datasets | | Stance | | Hate | | Hash | |
|---|---|---|---|---|---|---|---|
| Continued Pretraining Models | | UDALM | DPT | UDALM | DPT | UDALM | DPT |
| Retrievers ($N$=10) | RANDOM | $0.8043_{\pm0.6828}$ | $0.8052_{\pm0.7133}$ | $0.6615_{\pm0.7836}$ | $0.6604_{\pm0.5188}$ | $0.8209_{\pm3.746}$ | $0.8217_{\pm2.883}$ |
| | DPR | $0.8127_{\pm0.7066}$ | $0.8109_{\pm1.1388}$ | $0.6708_{\pm0.3764}$ | $0.6732_{\pm0.8230}$ | $0.8283_{\pm2.3412}$ | $0.8313_{\pm3.4901}$ |
| | BM25 | $0.8100_{\pm0.5595}$ | $0.8135_{\pm0.7299}$ | $0.6696_{\pm0.4929}$ | $0.6724_{\pm0.8503}$ | $0.8251_{\pm2.6470}$ | $0.8294_{\pm2.5524}$ |
| | Spider | $0.8105_{\pm0.5892}$ | $0.8112_{\pm0.3582}$ | $0.6704_{\pm0.2831}$ | $0.6663_{\pm0.7976}$ | $0.8246_{\pm3.5220}$ | $0.8275_{\pm3.1393}$ |
| | Condense | $0.8099_{\pm0.9194}$ | $0.8076_{\pm0.6605}$ | $0.6709_{\pm1.2127}$ | $0.6681_{\pm0.8108}$ | $0.8262_{\pm2.5812}$ | $0.8289_{\pm3.0780}$ |
| ERM (*baseline*) | | $0.8014_{\pm0.4353}$ (+0.0%) | | $0.6603_{\pm2.2869}$ (+0.0%) | | $0.8153_{\pm4.5249}$ (+0.0%) | |
| DPT (*SOTA*, $N$=300) | | $0.8247_{\pm0.5017}$ (+2.9%) | | $0.6734_{\pm1.0320}$ (+2.0%) | | $0.8301_{\pm7.8894}$ (+1.8%) | |
| **VIBE** ($N$=10) | | $\mathbf{0.8422_{\pm1.0150}}$ **(+5.1%)** | | $\mathbf{0.6950_{\pm1.0545}}$ **(+5.3%)** | | $\mathbf{0.8617_{\pm4.1626}}$ **(+5.7%)** | |

Table 3: Classification accuracy on retrieved adaptive data. Percentages in brackets indicate the relative increase compared to ERM. $N$ values mean we retrieve the top-$N$ retrieved data as adaptive data. VIBE outperforms all comparisons significantly (measured by the paired t-test with p-value$< 0.01$) and achieves an increase of more than 5% for three datasets, using adaptive data scale 3% of the DPT (the SOTA based on continued pretraining).

We then quantify the distribution shifts of training, test, and retrieved adaptive data with the commonly-used MMD criterion for measuring the feature difference (higher scores indicate a larger gap) (Gretton et al., 2012). In Figure 4, the MMD score of "stance: retrieve-test" is higher than that of "stance-hate: train". It means the distribution gap between the retrieved adaptive data and test data is even larger than the cross-task, cross-dataset distribution gap between stance and hate dataset.

**Baselines.** We first employ the ERM (Vapnik and Vapnik, 1998) baseline trained with past data only. Then, the **Domain Adaptation** (DA) baselines — adversarial-based DANN (Ganin et al., 2016), feature-consistency-regularization based Mixup (Yan et al., 2020), distribution-aligning based MMD (Li et al., 2018), and self-training based P+CFd (Ye et al., 2020) are compared. Besides, we consider state-of-the-art (SOTA) **Continued Pretraining** methods: UDALM (Karouzos et al., 2021) and DPT (Luu et al., 2022). To acquire large-scale adaptive data, we adopt **Retrieval** methods via RANDOM, DPR (Karpukhin et al., 2020), BM25 (Robertson et al., 2009), Spider (Ram et al., 2022), and Condense (Gao and Callan, 2021). All models' implementation details are shown in A.4.

## 6 Experimental Results

In this section, we first discuss the main results (§6.1). Then, an ablation study (§6.2) discusses the contributions of various VIBE modules. Next, we quantify VIBE's sensitivity to adaptive data scales in §6.3. Lastly, a case study (§6.4) interprets the features learned by VIBE in temporal adaptation.

### 6.1 Main Results

We first compare classification accuracy with golden adaptive data and choose the best baselines to experiment with retrieved adaptive data.

**Results on Golden Adaptive Data.** As shown in Table 2, we draw the observations as follows:

*Performances of all DA baselines are not comparative to the ERM, which minimizes the prediction losses on past training data only.* It echos findings of previous studies (Rosenfeld et al., 2022; Ye et al., 2021) with two-fold reasons. First, DA methods tend to minimize the distribution shift or extract the domain-invariant features instead of adapting to the temporal shift. Second, adapting to temporal distribution shifts among noisy data presents a challenge unable to be addressed by DA baselines.

*Continued pretraining methods are effective while VIBE performs the best.* The better results from UDALM and DPT than other DA baselines indicate that continued pretraining on high-quality data may partially help. They are outperformed by VIBE, implying that VIBE can better utilize the adaptive data for temporal adaptation by modeling topic evolution. Nevertheless, it is concerned the small data scales may limit continue-pretraining's performance, so we will then discuss how they work with large-scale retrieved adaptive data.

**Results on Retrieved Adaptive Data.** We have demonstrated VIBE's effectiveness with golden adaptive data split from the original dataset. However, in the real world, well-collected, high-quality data may be unavailable. Therefore, we explore the potential of utilizing retrieved adaptive data from online Twitter streams. We compare VIBE (using DPR retrieved data) to UDALM and DPT, the SOTA continued pretraining methods, with data collected by varying retrievers. The results are shown in Table 3, and we observe the following:

*Data quantity matters more to continued pretraining than quality.* UDALM and DPT perform comparably. For them both, retrieval slightly helps, yet the selection of retrievers barely affects their results. Meanwhile, scaling up the data quantity

| Dataset | | Stance |
| --- | --- | --- |
| Module Ablation | Vanilla NTM | $0.8200_{\pm 0.1250}$ |
| | IB-NTM | $0.8315_{\pm 0.1872}$ |
| | VIBE (Full) | $0.8422_{\pm 0.4995}$ |
| Topic Number ($K$) | 16 | $0.8213_{\pm 0.5803}$ |
| | 32 | $0.8286_{\pm 0.4538}$ |
| | 64 | $0.8379_{\pm 0.4782}$ |
| | 128 | $0.8422_{\pm 0.2234}$ |
| | 256 | $0.8361_{\pm 0.3423}$ |
| | 512 | $0.8340_{\pm 0.1867}$ |
| Retriever ($N$=10) | DPR | $0.8422_{\pm 0.2036}$ |
| | Spider | $0.8352_{\pm 0.2796}$ |
| | BM25 | $0.8309_{\pm 0.5810}$ |

Table 4: Ablation experimental results. Vanilla NTM: simple concatenation with BERT and NTM features. IB-NTM: VIBE without topic space projection (§4.5).

substantially boost DPT's results. This observation is consistent to Luu et al. (2022) that continued pre-training relies on large-scale data to work well.

*VIBE exhibits the largest performance gain compared to baseline (ERM).* VIBE uses 3% of the retrieved data while outperforming DPT with million-scale data. It implies that VIBE can utilize the data efficiently and its learned topic evolution is useful.

## 6.2 Ablation Study

We have shown the overall superiority of VIBE. To provide more insight, we further discuss it with *Stance* (other datasets exhibit similar trends). Here, we investigate the efficacy of its modules through the experimental ablation results in Table 4.

First, we compare module ablation results and find that either IB or topic-space projection contributes positively to VIBE, with their joint effects leading to the best results. Then we quantify the effects of the NTM's topic number ($K$) and observe first increase-then decrease trends. This is because small $K$ may cause underfitting, while large $K$ may lead to sparse topics, both hindering VIBE's capability of capturing high-quality topics. Finally, we examine the VIBE with varying retrievers and observe that DPR works the best in finding suitable adaptive data thanks to its large-scale pretraining.

In the second training stage of VIBE, future tweets are pseudo-labeled by the classifier trained in the first training stage. Thus we further explore the effects of label distribution of pseudo labels on model performance. We compare prediction accuracy on *Stance* with either ground-truth pseudo labels or false pseudo labels and observe the accuracy of 84.72% and 83.06% respectively. While pseudo-labeling in self-training can introduce errors, which can potentially undermine its effectiveness, our findings suggest that when combined with posi-

tive samples, pseudo-labeling still yields promising results. In fact, the performance of pseudo-labeling is only marginally worse than the upper-bound results achieved with correctly-labeled data alone.

## 6.3 Effects of Adaptive Data Scale

As shown in §6.1, adaptive data scale ($N$) is crucial for DPT. We are then interested in how it affects VIBE and show the comparison of VIBE and DPT in Figure 5. As shown, VIBE consistently performs better and exhibits much less sensitivity to adaptive data quantity than DPT, where the former peaks with very small $N$ and the latter grows slowly.

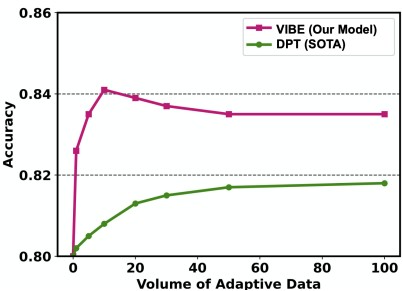

Figure 5: Accuracy (y-axis) over varying $N$ (adaptive data with the top-$N$ retrieved samples). VIBE shows a much smaller reliance on adaptive data scales than DPT.

## 6.4 Case Study on Evolving Topic Patterns

To further examine how VIBE works, we qualitatively analyze the learned topics on Stance, where the changing stance towards Dr. *Anthony Fauci* is analyzed. We examine the stance labels of all tweets containing the word "Fauci" and find a rising number of tweets in favor of Fauci over time. By taking a closer look at these tweets, we find that word "Fauci" often co-occurs with "Trump", where specifically, in the tweets against Trump, people tend to support Fauci. In other words, the "usually-against" stance gained from "Trump"-tweets might, in return signal the "favor" stance for "Fauci" in

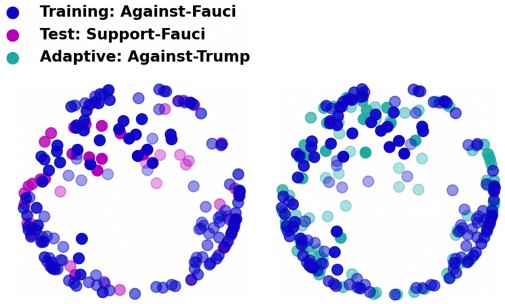

Figure 6: Tweets distribution in sphere topic space. Blue dots: against-Fauci tweets in past training data; Pink dots: support-Fauci tweets in future test data; Green dots: against-Trump tweets in adaptive data.

tweets where they are both discussed. Interestingly, we do observe a rising "for-Fauci" stance rate over time in tweets mentioning both Trump and Fauci.

We then visualize VIBE's sphere topic space of Fauci- and Trump-related samples in Figure 6. Recall that VIBE projects the reconstructed BoW vectors of tweets into a sphere space via multi-task training on predicting time and class labels to obtain time-aware coordinates of tweets (as shown in 4.5). In the left sphere, the distribution of against- and support-Fauci tweets from the past and future are different, which presents challenges for the model to infer the changing stance. Nevertheless, in the right sphere, the against-Trump tweets from adaptive data exhibit a similar distribution with support-Fauci tweets from future test data, allowing VIBE to adapt to the change. These observations show VIBE can effectively capture topic evolution, largely benefiting temporal adaptation.

## 7 Discussion on Application Scenarios

In this section, we envision more application scenarios where our model VIBE can potentially be applied for dynamic language understanding in a temporally evolving environment.

In the real world, the rapid changes in language environments challenge language models. Previous approaches often resort to continuously updating models to cope with shifted distributions and newly emerging knowledge, resulting in extensive resource consumption. Then how about taking advantage of large language models that are highly knowledgeable? We have explored applying a zero-shot large language model (LLM) LLaMa (Touvron et al., 2023), while exhibiting compromised performance compared with the transformer-based VIBE, showing that purely scaled-up language models can not tackle temporal misalignment. Moreover, it may be prohibitively expensive to keep updating LLM's knowledge via continual training, and the challenge will be exacerbated as models continue to scale up (Bender et al., 2021; Ziems et al., 2023).

Inspired by that topic evolution can indicate language evolution in a temporally changing environment (Zhang and Li, 2022), VIBE unveils a low-cost temporal adaptation approach by reflecting on past and future topic statistics simultaneously to infer evolution, which is efficient and practical in the real world. VIBE is built upon the general text classification task, and it encodes sentence-level temporal semantics, hence having the potential to generalize over more complex tasks with varying data resources (Zeng et al., 2018; Sun et al., 2023).

Many social media applications suffer from temporal misalignment caused by distribution shifts over time. For instance, in a dynamic environment, how to recommend users items that align with their evolving interests (Zhang et al., 2021)? Furthermore, harmful texts may contain new semantics over time, making non-adapted models fail to detect them. In the context of rumor detection, as rumors are tested by facts over time, learning new facts to reinforce rumor detection is necessary. More language understanding tasks also face the challenge of comprehending changing texts in dynamic language environments, resulting in compromised end-task performance, which is explored by Zhang and Li (2022)'s empirical study on the effects of temporal misalignment on social media downstream tasks when there is a time gap between training and test data. We will explore our model VIBE in more application scenarios like the aforementioned tasks in the future to enhance performance of language models in the real world.

## 8 Conclusion

We have presented the first study on the effects of latent topic evolution for temporal adaptation in noisy contexts. A novel model VIBE has been proposed leveraging the information bottleneck to capture topic evolution. In experiments, VIBE significantly outperforms the continued pretraining methods with much smaller scales of adaptive data.

## Limitations

In the following, we summarize limitations of our study, which should be considered for future work.

For the data we employ for adaptation, the retrieved adaptive data was first crawled online and then retrieved by DPR. In the ablation study, we analyzed the difference of employing different retrievers to retrieve future data for adaptation. The results demonstrate that although adaptive data collected by different retrievers all benefit temporal adaptation, the four retrievers exhibit varying influences on the quality of adaptive data, which indicates the requirement for retrievers to select suitable adaptive data. Thus the DPR should be continued-trained in the future to keep pace with the evolving online environments.

For the research scope, our paper focuses on temporal adaptation for text classification on so-

cial media data. In the future, we will broaden our research scope to more applications like recommendation or generalization on dynamic contexts.

## Ethics Statement

In our empirical study, datasets Stance and Hate are publicly available with well-reputed previous work. These two datasets in general does not create direct societal consequence. The stance detection dataset helps us understand how the public opinions changed during the COVID-19, shedding lights for future work on public opinions study. The hate speech dataset is for defending against hate speech by training models to detect and prevent them. Our model aims to enhance model performance to better detect hate speech and eliminate them in evolving environments, which is for social benefit. The Stance and Hate datasets are for research purpose only. For experimental results, all the qualitative results we discuss are output by machine learning models, which do not represent the authors' personal views. Meanwhile, user information was excluded from these two datasets.

For the data collection of Hash dataset, Twitter's official API was employed strictly following the Twitter terms of use. The newly gathered data was thoroughly examined to exclude any possible ethical risks, e.g., toxic language and user privacy. We also conducted data anonymization in pre-processing by removing user identities and replacing @mention with a generic tag. We ran a similar process for adaptive data's auto-collection.

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

# A  Appendix

## A.1  Vanilla Neural Topic Model

In this section, we elaborate on the vanilla neural topic model, which is the base of our model VIBE.

The potential of NTM to deal with dynamics comes from its capability of clustering posts exhibiting similar word statistics and forming latent topics to reflect their shared discussion point. Therefore, the intra-cluster content, though varying in the generation time, reflects the implicit semantic consistency throughout time and enables the learning of underlying past-to-future connections.

The NTM includes an encoder and a decoder. For the input, a tweet $\mathbf{t}$ is firstly mapped into the bag-of-word (BoW) vector form, then fed into the auto-encoder. Given the BoW input $\mathbf{t}$, the clustering is conducted through auto-encoding, which contains an encoding process to map $\mathbf{t}$ into a latent topic vetcor $\mathbf{z}_t$, followed by decoding to rebuild $\mathbf{t}$ conditioned on the topic ($\mathbf{z}_t$). During this process, $\mathbf{z}_t$ is a $K$ dimensional vector; each entry reflects the chance of $\mathbf{t}$ clustering into a certain topic and $K$ is a hyper-parameter representing the total topic number in corpus. Below presents concrete steps.

For encoding, $\mathbf{t}$ is embedded into the latent topic space to generate $\mathbf{z}_t$ via Gaussian sampling, where the mean $\mu$ and standard deviation $\sigma$ are learned with the following formula:

$$\mu = f_\mu(f_e(\mathbf{t})), \log \sigma = f_\sigma(f_e(\mathbf{t})) \tag{20}$$

$f_*(\cdot)$ is a ReLU-activated neural perceptron. Then $\mathbf{z}_t$ is drawn from the normal distribution below:

$$\mathbf{z}_t = \mathcal{N}(\mu, \sigma) \tag{21}$$

It is later transformed to a distributional vector via a softmax function to yield $\theta_t$, representing the topic mixture of $\mathbf{t}$. It initiates decoding step to re-construct $\mathbf{t}$ by predicting $\hat{\mathbf{t}}$ below:

$$\hat{\mathbf{t}} = softmax(f_\phi(\theta_t)) \tag{22}$$

$f_\phi(\cdot)$ is another ReLU-activated perceptron mapping information in topic space back to the BoW. The weights of $f_\phi(\cdot)$ (after softmax normalization) are employed to represent the topic-word distributions and the latent topic vector $\mathbf{z}_t$ (with cross-time views gained in clustering) can be engaged in classification (Eq. 15) to capture dynamics over time.

## A.2  Proofs

In this section, we explain some derivations from our formulations in detail.

**ELBO for $\mathbf{p}_\theta(\mathbf{t^x}, \mathbf{t^y})$.** We explain how the ELBO of $p_\theta(t^x, t^y)$ is derived, as shown in Eq. 23.

**IB Regularization on Mutual Information.** We explain how Eq. 5-Eq. 7 are derived here. Interaction Information between X, Y, and Z is:

$$
\begin{aligned}
I(X;Y;Z) &= I(X;Y) - I(X;Y|Z) \\
&= I(X;Z) - I(X;Z|Y) \\
&= I(Y;Z) - I(Y;Z|X)
\end{aligned}
\tag{24}
$$

$$\log p(t^x, t^y) = \log \int p\left(t^x \mid z^x, z^s\right) p\left(t^y \mid z^y, z^s\right) p\left(z^x\right) p\left(z^s\right) p\left(z^y\right) dz^x dz^s dz^y$$

$$= \log \int \frac{p\left(t^x \mid z^x, z^s\right) p\left(t^y \mid z^y, z^s\right) p\left(z^x\right) p\left(z^s\right) p\left(z^y\right)}{q\left(z^x \mid t^x\right) q\left(z^s \mid t^x, t^y\right) q\left(z^y \mid t^y\right)} q\left(z^x \mid t^x\right) q\left(z^s \mid t^x, t^y\right) q\left(z^y \mid t^y\right) dz^x dz^s dz^y$$

$$= \log \mathbb{E}_{q(z^x \mid t^x) q(z^s \mid t^x, t^y) q(z^y \mid t^y)} \left[ \frac{p\left(t^x \mid z^x, z^s\right) p\left(t^y \mid z^y, z^s\right) p\left(z^x\right) p\left(z^s\right) p\left(z^y\right)}{q\left(z^x \mid t^x\right) q\left(z^s \mid t^x, t^y\right) q\left(z^y \mid t^y\right)} \right] \qquad (23)$$

$$\geq \mathbb{E}_{q(z^x \mid t^x) q(z^s \mid t^x, t^y) q(z^y \mid t^y)} \left[ \log \frac{p\left(t^x \mid z^x, z^s\right) p\left(t^y \mid z^y, z^s\right) p\left(z^x\right) p\left(z^s\right) p\left(z^y\right)}{q\left(z^x \mid t^x\right) q\left(z^s \mid t^x, t^y\right) q\left(z^y \mid t^y\right)} \right]$$

$$= \mathbb{E}_{q(z^x \mid t^x) q(z^s \mid t^x, t^y)} \left[ \log p\left(t^x \mid z^x, z^s\right) \right] + \mathbb{E}_{(z^s \mid t^x, t^y) q(z^y \mid t^y)} \left[ \log p\left(t^y \mid z^y, z^s\right) \right]$$

$$- D_{KL} \left[ q\left(z^x \mid t^x\right) \| p\left(z^x\right) \right] - D_{KL} \left[ q\left(z^s \mid t^x, t^y\right) \| p\left(z^s\right) \right] - D_{KL} \left[ q\left(z^y \mid t^y\right) \| p\left(z^y\right) \right]$$

Eq. 5 and Eq. 6 are derived similarly. Moreover, given $I(X;Z) - I(X;Z|Y) = I(Y;Z) - I(Y;Z|X)$, the mutual information between $Z^X$ and $Z^Y$ is:

$$I(Z^X; Z^S) = I(Z^X; X) - I(Z^X; X|Z^S) + I(Z^X; Z^S|X) \qquad (25)$$

Since $q(z^x|x) = q(z^x|x, z^s)$, the last term in Eq. 25 equals zero, whose calculation goes as:

$$I(Z^X; Z^S|X) = H(Z^X|X) - H(Z^X|X, Z^S)$$
$$= H(Z^X|X) - H(Z^X|X) = 0 \qquad (26)$$

Thus we eliminate the zero term and get:

$$I(Z^X; Z^S) = I(Z^X; X) - I(Z^X; X|Z^S) \qquad (27)$$

As Eq. 24, $-I(Y;Z|X) = I(Y;Z) - I(X;Y;Z)$, similarly $-I(Z^X; X|Z^S) = I(X; Z^S) - I(X; Z^X; Z^S)$. Thus:

$$I(Z^X; Z^S) = I(X; Z^X) + I(X; Z^S - I(X; Z^X, Z^S)). \qquad (28)$$

**Tractaility for Joint Regularization.** Hwang et al. (2020) elaborates how the lower bounds of $I(X; Z^X, Z^S)$, $I(X; Z^S|Y)$ and $I(X; Z^X)$ are derived detailedly via variational approximation.

## A.3 Dataset

For the three datasets, we adopted the Twitter API for recovery[1] of tweets with missing information (e.g., the timestamp). The Stance detection dataset contains tweets with annotated stances about various COVID-19 topics (Glandt et al., 2021). The Hate Speech data was released by Mathew et al. (2021) with labels indicating whether or not the hate speech exists in a tweet. These two datasets are from publicly available benchmarks with relatively clean annotations. However, many social media applications are built upon noisy user-generated labels (Wang et al., 2019; Zhang et al., 2021).

Thus we built Hash dataset for studying the user-generated labels instead of well-annotated clean data. In this dataset, the classification label is a hashtag the author annotates to indicate a tweet's topic. Its data was gathered following Nguyen et al. (2020) with English tweets posted from September to December 2011, to allow a balanced year coverage across different datasets. Following Zhang et al. (2021), the top 10 hashtags with the highest frequency were selected to be the labels. Then tweets containing these hashtag labels were preserved to group the Hash dataset.

## A.4 Implementation Details

We discuss the implementation details of our model VIBE and comparison baselines in this section.

For **retrievers**, we adopted the DPR[2] model of the Natural Questions BERT Base checkpoint (Karpukhin et al., 2020). This model has been pretrained on the Natural Questions dataset (Kwiatkowski et al., 2019) for open-domain question answer. For other retrievers, we also adopted their pretrained checkpoint for retrieval.

For **domain adaptation** baselines, we employed the official implementation[3] of them. Since these domain adaptation baselines were implemented on image processing tasks, we modified the part of the image processing code to natural language processing code. We trained each model for 40 epochs and fixed parameters via grid search on validation data.

For **continued pretraining**, UDALM fine-tunes itself using a mixed classification and Masked Language Model loss on past training data and future adaptive data together. For DPT, we used unlabeled future adaptive data for continued pretraining of BERT with MLM loss and then fine-tuned the re-

---

[1] https://developer.twitter.com/en/docs/twitter-api

[2] https://github.com/facebookresearch/DPR

[3] https://github.com/facebookresearch/DomainBed

sulting model on labeled past training data. We implemented the two based on official code[4] for BERT (base, uncased). UDALM was trained for 40 epochs. DPT was trained for 30 epochs for pretraining and 10 epochs for fine-tuning.

For our model **VIBE**, we took the BERT (base, uncased) for the BERT encoding, to ensure a fair comparison with the continued pretraining baselines. For all the MLPs, the hidden layer size was set to 2048. For NTM, the topic number for each latent topic including $z^x$, $z^y$, and $z^s$ is set to 128. The corpus size is set to 20,000. VIBE is trained in two stages. For the first stage of training, The $\mathcal{L}_{ntm}$ in Eq. 13 is first optimized via updating NTM parameters for one epoch for warm-up, then $\mathcal{L}_{past}+\mathcal{L}_{NTM}$ are jointly optimized via updating NTM, BERT, and MLP parameters for 10 epochs. The parameters are fixed via grid search on validation data. Then the pretrained classifier was used to pseudo-label the unlabeled adaptive data. Then VIBE was trained in second stage, where the reconstructed BoW vectors of past and future tweets were projected to the sphere topic space via multi-task training on inferring time and class labels simultaneously. During this, all parameters of VIBE were updated together and fixed via grid search on validation data for 10 epochs.

---

[4]https://github.com/huggingface/transformers