# OpenReview forum: "VIBE: Topic-Driven Temporal Adaptation for Twitter Classification"
_EMNLP/2023/Conference — EMNLP 2023 Main_

### Official Review · Reviewer_7trR · 2023-07-22

**Soundness:** 5

**Excitement:**

4: Strong: This paper deepens the understanding of some phenomenon or lowers the barriers to an existing research direction.

**Paper Topic And Main Contributions:**

This paper studies temporal adaptation, where models trained on past data are tested in the future. To do so, they propose a new model, VIBE: Variational Information Bottleneck for Evolutions. Specifically, they first employ two Information Bottleneck (IB) regularizers to distinguish past and future topics. Then, the distinguished topics work as adaptive features via multi-task training with the timestamp and class label prediction. In adaptive learning, VIBE utilizes retrieved unlabeled data from Twitter streams created posterior to training data time.

**Reasons To Accept:**

1. They propose a new task which would be beneficial to researchers working on temporal information extraction and topic modeling.
2. They have conducted extensive experiments on three datasets, which makes their conclusions solid and sound. Additionally, the detailed ablation study and case study also provide insights into the future research direction.

**Reasons To Reject:**

1. All three datasets they used are tweets. It would be beneficial to conduct experiments on some formal text datasets, such as news and reports.
2. The application scenarios should be included.

**Reproducibility:**

4: Could mostly reproduce the results, but there may be some variation because of sample variance or minor variations in their interpretation of the protocol or method.

**Reviewer Confidence:**

4: Quite sure. I tried to check the important points carefully. It's unlikely, though conceivable, that I missed something that should affect my ratings.

---

> ### Author Rebuttal · Authors · 2023-08-28
>
> Thanks for your insightful comments!
>
> Q4.1: Why choose Twitter datasets for experiments, how about more datasets like news and reports?
>
> A4.1: Thanks! This paper’s scope is tweet classification because Twitter data presents a unique challenge of data noisiness in temporally evolving environments. Twitter data contains millions of varying topics discussed by the public, which change very fast as time goes by, making it challenging for models to keep pace with the content evolution (as discussed by our paper in lines 449-475). The rapidly evolving Twitter data reflects a realistic scenario for temporal adaptation study. For further study, we will expand our research scope to more applications like news and reports in the future.
>
> Q4.2: The application scenarios should be included.
>
> A4.2: Thanks! Many real-world applications are challenged by the evolving language features in social media platforms. For example, as time passes, the hate speech detection classifier may fail to recognize some new cases of hate speech due to the outdated training data, our method can adapt the outdated hate speech detectors to the future, contributing to a higher detection accuracy of new hate speech. Additionally, our approach can be generalized to more scenarios like making accurate recommendations for users when facing changing environments. We will add more discussion on application scenarios in the revised version.

---

### Official Review · Reviewer_vJ1w · 2023-07-31

**Soundness:** 4

**Excitement:**

4: Strong: This paper deepens the understanding of some phenomenon or lowers the barriers to an existing research direction.

**Paper Topic And Main Contributions:**

The paper introduces VIBE, a model designed to address the challenge of deteriorating text classification performance in dynamic real-world social media. VIBE utilizes two Information Bottleneck regularizers to distinguish past and future topics, enabling it to adapt to changing language features. Through multi-task training and leveraging unlabeled data from online streams, VIBE significantly outperforms previous state-of-the-art continued pretraining methods with 3% of the data in experiments on Twitter for three classification tasks.

**Reasons To Accept:**

1.	This paper introduces an approach to address the problem of deteriorating text classification performance in dynamic real-world social media. This innovative perspective allows the model to adapt effectively to changing language features over time, improving performance.
2.	VIBE leverages two Information Bottleneck regularizers to distinguish past and future topics. Using these distinguished topics as adaptive features and incorporating unlabeled data from online streams, the model achieves significant performance gains in text classification tasks on Twitter data.
3.	The paper presents substantial experiments conducted on real-world social media data for three different text classification tasks. The results demonstrate the superior performance of VIBE.

**Reasons To Reject:**

1.	The model looks like a complex solution to a simple text classification task.
2.	Do the authors think that when the language model scale is large enough, it can ignore the impact of data temporal differences on classification performance? Why didn't the authors try to use large language models like GPT3.5 as their baseline models?
3.	How are the labels distributed in each dataset? Why did the author only apply accuracy as an evaluation metric?

**Reproducibility:**

4: Could mostly reproduce the results, but there may be some variation because of sample variance or minor variations in their interpretation of the protocol or method.

**Reviewer Confidence:**

3: Pretty sure, but there's a chance I missed something. Although I have a good feel for this area in general, I did not carefully check the paper's details, e.g., the math, experimental design, or novelty.

---

> ### Author Rebuttal · Authors · 2023-08-28
>
> Thanks for your insightful comments!
>
> Q3.1: Why does this paper propose a complex solution?
>
> A3.1: Thanks! Our paper addresses the challenging research problem of the deteriorating model performance over time in real-world applications. This issue is more of a temporal adaptation task[3, 4, 5, 6] than a text classification task. Our model design is necessary as it is challenging to capture the rapid evolution of language features in social media environments as time progresses. Previous studies on domain adaptation and pretraining baselines all exhibited compromised performance when faced with changing environments, highlighting the urgent need for a more advanced framework. Additionally, we follow the previous temporal adaptation benchmark [3, 4] to experiment with various text classification tasks and examine the capability of NLU models to comprehend temporal dynamics. In the future, we plan to explore temporal adaptation in more complex scenarios, such as text generation.
>
> Q3.2: Can scaling up language models like GPT3.5 tackle the temporal influences on models? How about adapting GPT3.5 to the changing future?
>
> A3.2: Thanks! In recent papers [7, 8, 9], it has been shown that the scaled-up language models exhibit compromised performance on out-of-distribution data. This highlights the inferiority of directly using extra-large language models to solve the temporal adaptation problem, which is especially challenging when faced with rapid, synchronous analysis of breaking events. Updating LLM’s knowledge of temporal events may be prohibitively expensive via continued pretraining [10], and this challenge will only be exacerbated as models continue to scale up. Even the official publisher for GPT3.5 has not updated it since 2021, which further demonstrates the impracticality of adapting these models for researchers who have limited computational resources.
>
> While continued pretraining GPT3.5 for temporal adaptation is extremely expensive, we experimented on continued pretraining BERT. However, we observed that this approach still requires a significant amount of training samples and computational resources, yet it exhibits lower-than-expected performance (as shown in Table 3). In contrast, our proposed model VIBE offers a compelling alternative. It can adapt models to the future and achieve state-of-the-art performance at a much lower cost. With only 3% of the data used in continued pretraining, VIBE shows promising potential for addressing temporal adaptation challenges in the rapidly changing real world. For more interpretation, we will report the performance of extra-large language models in the revised version.
>
> Q3.3: Why adopt accuracy as the evaluation metric?
>
> A3.3: Thanks! For the evaluation metric, we follow previous work’s practice of utilizing prediction accuracy in evaluating model performance in the temporal setting, which directly reflects the model’s ability to adapt to changing data [11, 12]. To provide more insights, We will add the label distribution analysis in the revised version.
>
> References:
>
> [3] Kelvin Luu, Daniel Khashabi, Suchin Gururangan, Karishma Mandyam, and Noah A. Smith. 2022. Time Waits for No One! Analysis and Challenges of Temporal Misalignment. In Proceedings of the 2022 Conference of the North American Chapter of the Association for Computational Linguistics: Human Language Technologies, pages 5944–5958, Seattle, United States. Association for Computational Linguistics.
>
> [4] Paul Röttger and Janet Pierrehumbert. 2021. Temporal Adaptation of BERT and Performance on Downstream Document Classification: Insights from Social Media. In Findings of the Association for Computational Linguistics: EMNLP 2021, pages 2400–2412, Punta Cana, Dominican Republic. Association for Computational Linguistics.
>
> [5] Mourao, Fernando, Leonardo Rocha, Renata Araújo, Thierson Couto, Marcos Gonçalves, and Wagner Meira Jr. "Understanding temporal aspects in document classification." In Proceedings of the 2008 International Conference on Web Search and Data Mining, pp. 159-170. 2008.
>
> [6] Alkhalifa, Rabab, Elena Kochkina, and Arkaitz Zubiaga. "Building for tomorrow: Assessing the temporal persistence of text classifiers." Information Processing & Management 60, no. 2 (2023): 103200.
>
> [7] Linyi Yang, Shuibai Zhang, Libo Qin, Yafu Li, Yidong Wang, Hanmeng Liu, Jindong Wang, Xing Xie, and Yue Zhang. 2023. GLUE-X: Evaluating Natural Language Understanding Models from an Out-of-Distribution Generalization Perspective. In Findings of the Association for Computational Linguistics: ACL 2023, pages 12731–12750, Toronto, Canada. Association for Computational Linguistics.
>
> [8] Wang, Jindong, H. U. Xixu, Wenxin Hou, Hao Chen, Runkai Zheng, Yidong Wang, Linyi Yang et al. "On the Robustness of ChatGPT: An Adversarial and Out-of-distribution Perspective." In ICLR 2023 Workshop on Trustworthy and Reliable Large-Scale Machine Learning Models. 2023.
>
> [9] Kocoń, Jan, Igor Cichecki, Oliwier Kaszyca, Mateusz Kochanek, Dominika Szydło, Joanna Baran, Julita Bielaniewicz et al. "ChatGPT: Jack of all trades, master of none." Information Fusion (2023): 101861.
>
> [10] Bender, Emily M., Timnit Gebru, Angelina McMillan-Major, and Shmargaret Shmitchell. "On the dangers of stochastic parrots: Can language models be too big?🦜." In Proceedings of the 2021 ACM conference on fairness, accountability, and transparency, pp. 610-623. 2021.
>
> [11] Kasai, Jungo, Keisuke Sakaguchi, Yoichi Takahashi, Ronan Le Bras, Akari Asai, Xinyan Yu, Dragomir Radev, Noah A. Smith, Yejin Choi, and Kentaro Inui. "RealTime QA: What's the Answer Right Now?." arXiv preprint arXiv:2207.13332 (2022).
>
> [12] Zaporojets, Klim, Lucie-Aimée Kaffee, Johannes Deleu, Thomas Demeester, Chris Develder, and Isabelle Augenstein. "TempEL: Linking dynamically evolving and newly emerging entities." Advances in Neural Information Processing Systems 35 (2022): 1850-1866.

---

### Official Review · Reviewer_QHXk · 2023-08-05

**Soundness:** 4

**Excitement:**

4: Strong: This paper deepens the understanding of some phenomenon or lowers the barriers to an existing research direction.

**Paper Topic And Main Contributions:**

This paper proposes Variational Information Bottleneck for Evolutions (VIBE) for temporal adaptation in tweet classification tasks. VIBE models latent topic evolution by explicitly distinguishing past-exclusive, future-exclusive, and time-invariant latent topics via information bottleneck regularization. The authors further leverage multi-task learning to infer timestamps and class labels jointly to inject time-awareness senses into the classification training. The authors conduct experiments on three tweets classification datasets and demonstrate that VIBE outperforms state-of-the-art continued pertaining methods with a smaller amount of adaptive data. The authors also conduct comprehensive ablation studies to investigate the effectiveness of VIBE's different modules. A case study of changing stance towards Dr. Anthony Fauci is presented to demonstrate VIBE can effectively capture topic evolution.

**Reasons To Accept:**

1. The proposed method VIBE is effective. It outperforms the state-of-the-art continued pretraining methods with a smaller amount of adaptive data. The paper consists of a comprehensive set of preliminary analyses, classification experiments, ablation studies, and a case study to show the effectiveness of VIBE.

2. The authors conduct extensive theoretical derivation to obtain the information bottleneck regularization objectives.

3. The paper is clearly written and easy to follow.

**Reasons To Reject:**

1. The presentation of the paper can be improved.

**Reproducibility:**

4: Could mostly reproduce the results, but there may be some variation because of sample variance or minor variations in their interpretation of the protocol or method.

**Reviewer Confidence:**

3: Pretty sure, but there's a chance I missed something. Although I have a good feel for this area in general, I did not carefully check the paper's details, e.g., the math, experimental design, or novelty.

**Typos Grammar Style And Presentation Improvements:**

Line 267, Equation (3), $q_{\Phi_X}(z^s|t^x)$ -> $q_{\Phi_X}(z^x|t^x)$.

In the tables, I would suggest using a better way to show the standard deviation. Although the authors stated in Table 2 that the order of magnitude for std. is 10−5, it is still confusing to see a classification accuracy like 0.8109±1.1388.

---

> ### Author Rebuttal · Authors · 2023-08-28
>
> Thanks for your insightful comments!
>
> Q2.1: The presentation of the paper can be improved.
>
> A2.1: Thanks! We will refine the presentation of this paper in the revised version, including the equation and the standard deviation.

---

### Official Review · Reviewer_MQvq · 2023-08-06

**Typos Grammar Style And Presentation Improvements:** No grammatical mistakes I could find …
**Soundness:** 4

**Excitement:**

4: Strong: This paper deepens the understanding of some phenomenon or lowers the barriers to an existing research direction.

**Missing References:**

No missing reference as far as I know.

**Paper Topic And Main Contributions:**

The paper deals with the problem of Temporal Adaptation for Tweets classification. The authors model the latent topic evolution using a modified Neural Topic Model by learning time-dependent and time-invariant topics and later using the topics for label classification.
Authors adapt the Information Bottleneck principle to time-invariant and time-variant IB regularizers to learn the topics. The time-invariant representation is injected with the class information by training it to predict the class labels. Experiments on Tweet classification datasets demonstrate the efficacy of the proposed method over well-established baselines.

**Questions For The Authors:**

- Have the authors considered using a different input representation other than bag-of-words for the topic model learning?

**Reasons To Accept:**

- The paper is very well-written and easy to understand. I have not read topic model papers in some time, but I could readily follow the mathematical derivations and ideas.
- Authors use topic models to combine the latest transformer-based representations with traditional Bag-of-words (BoW) representations very elegantly in the proposed method. It's rare to see topic model papers, and exciting to see one.
- The experimental results are convincing; the authors compare the proposed method with domain adaptation baselines, with continued pretraining baselines covering multiple baselines that tackle the feature drift problem.

**Reasons To Reject:**

- I have no serious concerns with the paper, it's very well-written, proposes a solid technical contribution, and good empirical results.

**Reproducibility:**

4: Could mostly reproduce the results, but there may be some variation because of sample variance or minor variations in their interpretation of the protocol or method.

**Reviewer Confidence:**

4: Quite sure. I tried to check the important points carefully. It's unlikely, though conceivable, that I missed something that should affect my ratings.

---

> ### Author Rebuttal · Authors · 2023-08-28
>
> Thanks for your insightful comments!
>
> Q1.1: What about using a different input representation other than bag-of-words for the topic modeling?
>
> A1.1: Thanks! The reason for adopting the Bag-of-Words (BoW) representation is that it reflects word statistics from past to future and provides a sparse representation that complements the dense representation captured by transformer encoders. Previous studies have also demonstrated the advantages of using BoW representations in modeling noisy social media text [1, 2]. We will further explore different forms of representation for topic modeling in future research.
>
> References:
>
> [1] Dieng, Adji B., Francisco JR Ruiz, and David M. Blei. "Topic modeling in embedding spaces." Transactions of the Association for Computational Linguistics 8 (2020): 439-453.
>
> [2] Lixing Zhu, Gabriele Pergola, Lin Gui, Deyu Zhou, and Yulan He. 2021. Topic-Driven and Knowledge-Aware Transformer for Dialogue Emotion Detection. In Proceedings of the 59th Annual Meeting of the Association for Computational Linguistics and the 11th International Joint Conference on Natural Language Processing (Volume 1: Long Papers), pages 1571–1582, Online. Association for Computational Linguistics.

---

### Meta-Review · Area_Chair_vXZ4 · 2023-09-19

**Recommendation:** 5

**Metareview:**

The paper addresses the problem of temporal adaptation for tweet classification and introduces the Variational Information Bottleneck for Evolutions (VIBE) model. The paper explores temporal adaptation in tweet classification, which is a valuable perspective for handling dynamic social media data.

Most reviewers find the paper well-written, technically solid, and empirically convincing (MQvq and vJ1w). Reviewers have recognized the use of Information Bottleneck regularizers to distinguish past and future topics as an innovative approach to adapting to changing language features over time. The extensive experiments conducted on real-world social media data, along with ablation studies and case studies, are commended for demonstrating the effectiveness of the VIBE model. The clear and well-structured presentation of the paper, along with its readability, is appreciated by reviewers.

Reviewer QHXk suggests that the paper's presentation could be improved without specifying particular areas for enhancement. Reviewer vJ1w raises questions about the complexity of the VIBE model for text classification and suggests exploring the impact of data temporal differences on classification performance, especially in the context of larger language models (like GPT 3.5). More information is also asked on labels in each dataset and the choice of evaluation metrics. Additionally, conducting experiments on formal text datasets, such as news and reports, in addition to tweets, will be helpful (7trR).

In summary, the paper constitutes an innovative approach, comprehensive experiments, and a clear presentation. There are no major concerns with the paper's technical contributions. As suggested by reviewers, authors should consider including LLMs as a baseline, temporal differences in performance, and experiments on different text sources.

---

### Decision · Program_Chairs · 2023-10-07

**Decision:**

Accept-Main

**Comment:**

The paper addresses the problem of temporal adaptation for tweet classification and introduces the Variational Information Bottleneck for Evolutions (VIBE) model. The paper explores temporal adaptation in tweet classification, which is a valuable perspective for handling dynamic social media data.

Most reviewers find the paper well-written, technically solid, and empirically convincing (MQvq and vJ1w). Reviewers have recognized the use of Information Bottleneck regularizers to distinguish past and future topics as an innovative approach to adapting to changing language features over time. The extensive experiments conducted on real-world social media data, along with ablation studies and case studies, are commended for demonstrating the effectiveness of the VIBE model. The clear and well-structured presentation of the paper, along with its readability, is appreciated by reviewers.

Reviewer QHXk suggests that the paper's presentation could be improved without specifying particular areas for enhancement. Reviewer vJ1w raises questions about the complexity of the VIBE model for text classification and suggests exploring the impact of data temporal differences on classification performance, especially in the context of larger language models (like GPT 3.5). More information is also asked on labels in each dataset and the choice of evaluation metrics. Additionally, conducting experiments on formal text datasets, such as news and reports, in addition to tweets, will be helpful (7trR).

In summary, the paper constitutes an innovative approach, comprehensive experiments, and a clear presentation. There are no major concerns with the paper's technical contributions. As suggested by reviewers, authors should consider including LLMs as a baseline, temporal differences in performance, and experiments on different text sources.